# CONTINUOUS-FIDELITY BAYESIAN OPTIMIZATION WITH KNOWLEDGE GRADIENT

## ABSTRACT

While Bayesian optimization (BO) has achieved great success in optimizing expensive-to-evaluate black-box functions, especially tuning hyperparameters of neural networks, methods such as random search (Li et al., 2016) and multi-fidelity BO (e.g. Klein et al. (2017)) that exploit cheap approximations, e.g. training on a smaller training data or with fewer iterations, can outperform standard BO approaches that use only full-fidelity observations. In this paper, we propose a novel Bayesian optimization algorithm, the continuous-fidelity knowledge gradient (cfKG) method, that can be used when fidelity is controlled by one or more continuous settings such as training data size and the number of training iterations. cfKG characterizes the value of the information gained by sampling a point at a given fidelity, choosing to sample at the point and fidelity with the largest value per unit cost. Furthermore, cfKG can be generalized, following Wu et al. (2017), to settings where derivatives are available in the optimization process, e.g. large-scale kernel learning, and where more than one point can be evaluated simultaneously. Numerical experiments show that cfKG outperforms state-of-art algorithms when optimizing synthetic functions, tuning convolutional neural networks (CNNs) on CIFAR-10 and SVHN, and in large-scale kernel learning.

## 1 INTRODUCTION

In hyperparameter tuning of machine learning models, we seek to find a set of hyperparameters $x$ in some set $\mathbb{A}$ to minimize the validation error $f(x)$, i.e., to solve

$$\min_{x \in \mathbb{A}} f(x) \tag{1.1}$$

Evaluating $f(x)$ can take substantial time and computational power (Bergstra & Bengio, 2012), and may not provide gradient evaluations. Thus, machine learning practitioners have turned to Bayesian optimization for solving (1.1) (Snoek et al., 2012) because it tends to find good solutions with few function evaluations (Jones et al., 1998).

As the computational expense of training and testing a modern deep neural network for a single set of hyperparameters has grown as long as days or weeks, it has become natural to seek ways to solve (1.1) more quickly by supplanting some evaluations of $f(x)$ with computationally inexpensive low-fidelity approximations. Indeed, when training a neural network or most other machine learning models, we can approximate $f(x)$ by training on less than the full training data, or using fewer training iterations. Both of these controls on fidelity can be set to achieve either better accuracy or lower computational cost across a range of values reasonably modeled as continuous.

In this paper, we consider optimization with evaluations of multiple fidelities and costs where the fidelity is controlled by one or more continuous parameters. We model these evaluations by a real-valued function $g(x, s)$ where $f(x) := g(x, 1_m)$ and $s \in [0, 1]^m$ denotes the $m$ fidelity-control parameters. $g(x, s)$ can be evaluated, optionally with noise, at a cost that depends on $x$ and $s$. In the context of hyperparameter tuning, we may take $m = 2$ and let $g(x, s_1, s_2)$ denote the loss on the validation set when training using hyperparameters $x$ with a fraction $s_1$ of the training data and a fraction $s_2$ of some maximum allowed number of training iterations. We may also set $m = 1$ and let $s$ index either training data or training iterations. We assume $\mathbb{A}$ is a compact connected uncountable set into which it is easy to project, such as a hyperrectangle.

This problem setting also appears outside of hyperparameter tuning, in any application where the objective is expensive to evaluate and we may observe cheap low-fidelity approximations parameterized by a continuous vector. For example, when optimizing a system evaluated via a Monte Carlo simulator, we can evaluate a system configuration approximately by running with fewer replications. Also, when optimizing an engineering system modeled by a partial differential equation (PDE), we can evaluate a system configuration approximately by solving the PDE using a coarse grid.

Given this problem setting, we use the knowledge gradient approach (Frazier et al., 2009) to design an algorithm to adaptively select the hyperparameter configuration and fidelity to evaluate, to best support solving (1.1). By generalizing a computational technique based on the envelope theorem first developed in Wu et al. (2017), our algorithm supports parallel function evaluations, and also can take advantage of derivative observations when they are available. This algorithm chooses the point or set of points to evaluate next that maximizes the ratio of the value of information from evaluation against its cost.

Unlike most existing work on discrete- and continuous-fidelity Bayesian optimization, our approach considers the impact of our measurement on the future posterior distribution over the full feasible domain, while existing expected-improvement-based approaches consider its impact at only the point evaluated. One exception is the entropy-search-based method [10], which also considers the impact over the full posterior. Our approach differs from entropy search in that it chooses points to sample to directly minimize expected simple regret, while entropy search seeks to minimize the entropy of the location or value of the global optimizer, indirectly reducing simple regret.

We summarize our contributions as follows.

**Contributions of this paper:**

- We develop a novel Bayesian Optimization algorithm, the continuous-fidelity knowledge gradient (cfKG) method, which chooses the point and fidelity to sample next that maximizes the ratio of the value of information to its cost;

- After first developing this algorithm in the sequential derivative-free setting, and inspired by Wu et al. (2017), we generalize to settings where function evaluations may be performed in parallel, and where we can access possibly noisy and biased gradient information.

- We show that our algorithm outperforms a number of start-of-art benchmark algorithms when optimizing common synthetic functions, tuning convolutional neural networks on CIFAR-10 and SVHN, and in a large-scale kernel learning example.

The rest of the paper is organized as follows. Sect. 2 reviews related work. Sect. 3 presents the cfKG method. Sect. 4 tests cfKG on benchmarks including synthetic functions and hyperparameter tuning for deep learning and kernel learning. Sect. 5 concludes.

## 2 RELATED WORK

Algorithms exploiting inexpensive low-fidelity approximations for hyperparameter tuning have been proposed both within and outside the field of Bayesian optimization.

Outside of Bayesian optimization, Li et al. (2016) develops an early-stopping method called Hyperband that can outperform traditional Bayesian optimization when tuning hyperparameters within a random search framework by adaptively allocating a single predefined resource that can be taken to be the number of training iterations or the amount of training data. In contrast to Hyperband we allow more than one fidelity control, support derivative observations, and use one-step optimality in terms of value vs. cost to select the point or points to sample. We compare against Hyperband in our numerical experiments, and show improved performance training a convolutional neural network.

Using a modified expected improvement (EI) acquisition function, Huang et al. (2006) and Lam et al. (2015) develop methods for multi-fidelity settings with a finite discrete set of low-fidelity approximations. We call this the "discrete-fidelity setting" to contrast it with the continuous-fidelity setting we consider here. We do not compare against discrete-fidelity methods in our numerical experiments as both Kandasamy et al. (2017) and Klein et al. (2017) show that continuous-fidelity methods can find good solutions in continuous-fidelity problems as much as an order of magnitude faster than discrete-fidelity ones.

In the specific context of hyperparameter tuning, Domhan et al. (2015) proposes a simple early stopping criterion combined with SMAC (an EI acquisition function with a random forest based statistical model) in which evaluation of a configuration is stopped if its performance is predicted to be worse than the current best configuration.

Kandasamy et al. (2016) generalizes the upper confidence bound (UCB) criteria to the discrete-fidelity setting, and further generalizes to continuous fidelities in Kandasamy et al. (2017). We compare against Kandasamy et al. (2017) in our numerical experiments, finding that cfKG provides improved performance when the budget is sufficiently large. This may be because the UCB criteria was originally designed for minimizing the cumulative regret and not simple regret, with the simple regret bounds resulting from theoretical analysis of cumulative regret being loose enough to be consistent with the empirical performance gap we observe in our experiments. It may also be because both Kandasamy et al. (2016) and Kandasamy et al. (2017) use a two-stage process in which the point to evaluate is selected without considering the fidelity, and the fidelity only selecting afterward. In contrast, cfKG selects the point and fidelity jointly, and considers the impact of fidelity choice on the best point to sample.

Entropy search has also been extended to the discrete-fidelity setting by Swersky et al. (2013). Considering training iterations as a one-dimensional continuous fidelity control, Swersky et al. (2014) generalizes Swersky et al. (2013) to this specific continuous-fidelity setting. Klein et al. (2017) adapts entropy search to settings where the size of the training data is a one-dimensional continuous fidelity control. McLeod et al. (2017) extends predictive entropy search to settings with one fidelity control and provides a faster way to compute the acquisition function.

Poloczek et al. (2017) develops a knowledge gradient method for the discrete-fidelity setting, but does not consider the continuous-fidelity setting. Our computational techniques are quite different from the ones developed there, as necessitated by our consideration of continuous fidelities. As an advantageous byproduct, our computational techniques also avoid the need to discrete $\mathbb{A}$.

This paper proposes the first knowledge gradient method for continuous-fidelity settings, and show how to generalize it to the batch and derivative-enabled settings by generalizing the computational technique developed in Wu et al. (2017). To the best of our knowledge, this is the first multi-fidelity batch Bayesian optimization algorithm. The relationship of this paper to the existing multi-fidelity BO literature is summarized in Table. 1.

| Literature summary | EI / SMAC | Entropy Search | UCB | Knowledge Gradient |
|---|---|---|---|---|
| Discrete-fidelity | Huang et al. (2006); Lam et al. (2015) | Swersky et al. (2013) | Kandasamy et al. (2016) | Poloczek et al. (2017) |
| Continuous-fidelity | Domhan et al. (2015) | Swersky et al. (2014); Klein et al. (2017); McLeod et al. (2017) | Kandasamy et al. (2017) | **This paper** |

Table 1: Summary of the multi-fidelity BO literature.

Our method is also related to knowledge gradient methods for single-fidelity BO. In particular, the way in which we extend cfKG from the sequential (one evaluation at a time) derivative-free setting to allow batches of points and gradient observations follows work extending single-fidelity knowledge-gradient methods to the batch and derivative settings in Wu & Frazier (2016) and Wu et al. (2017). We also generalize the envelope-theorem based computational technique developed for the single-fidelity setting in Wu et al. (2017) to continuous fidelities. We compare with single-fidelity knowledge gradient methods in our numerical experiments, and show that cfKG is able to levarage inexpensive low-fidelity observations to provide improved performance in both derivative-free and derivative-enabled settings.

## 3    CONTINUOUS-FIDELITY KNOWLEDGE GRADIENT

In this section, we propose the continuous-fidelity knowledge gradient (cfKG), a novel Bayesian optimization algorithm that exploits inexpensive low-fidelity approximations. This algorithm, like most Bayesian optimization algorithms, consists of a statistical model and an acquisition function. cfKG proceeds iteratively by fitting the statistical model (described below in Section 3.1) to all

previously sampled (point, fidelty) pairs, and then finding and sampling the (point, fidelity) pair that maximizes the acquisition function. Once the budget is exhausted, at some iteration $N$, cfKG returns as its final solution the point with the lowest estimated $g(x, 1_m)$.

To describe cfKG in detail, Sect. 3.1 first describes Gaussian process regression for modeling both $g(x, s)$ and its cost of evaluation. This approach is standard, with the novelty arising in cfKG's acquisition function and how we optimize it. Then, Sect. 3.2 presents the cfKG acquisition function, which values sampling a (point, fidelity) pair according to the ratio of the value of the information gained from sampling that point at that fidelity, to the cost of doing so. Sect. 3.3 generalizes an envelope-theorem based computational technique developed in Wu et al. (2017) to efficiently maximize this acquisition function. We discuss extensions to the derivative-enabled setting in Sect. 3.4.

## 3.1 GAUSSIAN PROCESSES

We put a Gaussian process (GP) prior (Rasmussen & Williams, 2006) on the function $g$ or its logarithm. We describe this procedure placing the prior on $g$ directly, and then discuss below when we recommend instead placing it on $(x, s) \mapsto \log g(x, s)$. The GP prior is defined by its mean function $\mu^{(0)} : \mathbb{A} \times [0, 1]^m \mapsto \mathbb{R}$ and kernel function $K^{(0)} : \{\mathbb{A} \times [0, 1]^m\} \times \{\mathbb{A} \times [0, 1]^m\} \mapsto \mathbb{R}$. These mean and kernel functions have hyperparameters, whose inference we discuss below.

We assume that evaluations of $g(x, s)$ are subject to additive independent normally distributed noise with common variance $\sigma^2$. We treat the parameter $\sigma^2$ as a hyperparameter of our model, and also discuss its inference below. Our assumption of normally distributed noise with constant variance is common in the BO literature (Klein et al., 2017).

The posterior distribution of $g$ after $n$ function evaluations at points $z^{(1:n)} := \{(x^{(1)}, s^{(1)}), (x^{(2)}, s^{(2)}), \cdots, (x^{(n)}, s^{(n)})\}$ with observed values $y^{(1:n)} := \{y^{(1)}, y^{(2)}, \cdots, y^{(n)}\}$ remains a Gaussian process (Rasmussen & Williams, 2006), and $g \mid z^{(1:n)}, y^{(1:n)} \sim \text{GP}(\mu^n, K^{(n)})$ with $\mu^n$ and $K^{(n)}$ evaluated at a point $z = (x, s)$ (or pair of points $z, \tilde{z} = (\tilde{x}, \tilde{s})$) given as follows

$$\mu^{(n)}(z) = \mu^{(0)}(z) + K^{(0)}\left(z, z^{(1:n)}\right)\left(K^{(0)}(z^{(1:n)}, z^{(1:n)}) + \sigma^2 I\right)^{-1}\left(y^{(1:n)} - \mu(z^{(1:n)})\right),$$

$$K^{(n)}(z, \tilde{z}) = K^{(0)}(z, \tilde{z}) - K^{(0)}\left(z, z^{(1:n)}\right)\left(K^{(0)}(z^{(1:n)}, z^{(1:n)}) + \sigma^2 I\right)^{-1} K^{(0)}\left(z^{(1:n)}, \tilde{z}\right).$$
$$(3.1)$$

This statistical approach contains several hyperparameters: the variance $\sigma^2$, and any parameters in the mean and kernel functions. We treat these hyperparameters in a Bayesian way as proposed in Snoek et al. (2012).

When $g$ is the validation error in a hyperparameter optimization problem, we recommend putting a GP prior on $\log g(x, s)$, rather than on $g(x, s)$ directly, because (1) $g(x, s)$ is nonnegative and will be allowed to be negative after log scaling, better matching the range of values assumed by the GP, and (2) because $g(x, s)$ can climb steeply over several orders of magnitude as we move away from the optimal $x$, making $\log g(x, s)$ easier to model.

We analogously train a separate GP on the logarithm of the cost of evaluating $g(x, s)$.

## 3.2 THE CFKG ACQUISITION FUNCTION

cfKG samples the point and fidelity that jointly maximize an acquisition function, which we define in this section by adopting the knowledge gradient concept (Frazier et al., 2009) in the continuous-fidelity setting to value the information gained through one additional sample.

If we were to stop sampling after $n$ samples, we would select as our solution to (1.1) a point $x$ with minimum estimated validation error $\mu^{(n)}(x, 1_m)$, and this point would have a conditional expected validation error of $\min_{x \in \mathbb{A}} \mu^{(n)}(x, 1_m)$ under the posterior. If instead we took an additional sample at $x^{(n+1)}$ with the fidelity $s^{(n+1)}$, then the minimum expected validation error under the resulting posterior would become $\min_{x \in \mathbb{A}} \mu^{(n+1)}(x, 1_m)$. This quantity depends on $x^{(n+1)}$ and $s^{(n+1)}$ through the dependence of $\mu^{(n+1)}(x, 1_m)$ on the point and fidelity sampled, and is random under

the posterior at iteration $n$ because $\mu^{(n+1)}(x, 1_m)$ depends on the observation $y^{(n+1)}$. We discuss this dependence explicitly in Sect. 3.3.

The value of the information gained by sampling at $x^{(n+1)}$ with the fidelity $s^{(n+1)}$ conditioned on any particular outcome $y^{(n+1)}$ is thus the difference of these two expected validation errors $\min_{x \in \mathbb{A}} \mu^{(n)}(x, 1_m) - \min_{x \in \mathbb{A}} \mu^{(n+1)}(x, 1_m)$. We then take the expectation of this difference, over the random outcome $y^{(n+1)}$, to obtain the (unconditional) value of the information gained, and take the ratio of this value with the cost of obtaining it to obtain the cfKG acquistion function,

$$\text{cfKG}(x, s) = \frac{\min_{x' \in \mathbb{A}} \mu^{(n)}(x', 1_m) - \mathbb{E}_n \left[ \min_{x' \in \mathbb{A}} \mu^{(n+1)}(x', 1_m) \mid x^{(n+1)} = x, s^{(n+1)} = s \right]}{\text{cost}^{(n)}(x, s)}, \quad (3.2)$$

where $\text{cost}^{(n)}(x, s)$ is the estimated cost of evaluating at $x$ with the fidelity $s$ based on the observations available at iteration $n$, according to the GP described in Sect. 3.1, and $\mathbb{E}_n$ indicates the expectation taken with respect to the posterior given $x^{(1:n)}, s^{(1:n)}, y^{(1:n)}$.

The cfKG algorithm chooses to sample at the point and fidelity that jointly maximize the cfKG acquistion function

$$\max_{(x,s) \in \mathbb{A} \times [0,1]^m} \text{cfKG}(x, s). \quad (3.3)$$

Although this acquisition function considers the expected value of an improvement due to sampling, it differs from expected improvement approaches such as Lam et al. (2015) because the point at which an improvement occurs, $\arg\max_{x \in \mathbb{A}} \mu^{(n+1)}(x, 1_m)$ may differ from the point sampled. Moreover, this acquisition function allows joint valuation of both the point $x$ and the fidelity $s$, while approaches such as Lam et al. (2015) require valuing a point $x$ assuming it will be evaluated at full fidelity and then choose the fidelity in a second stage.

cfKG generalizes naturally to batch settings where we can evaluate multiple (point, fidelity) pairs at once. We value joint evaluation of $q \geq 1$ points $x_{1:q}$ at fidelities $s_{1:q}$, where $z_{1:q} = ((x_1, s_1), \ldots, (x_q, s_q))$, by

$$\text{q-cfKG}(z_{1:q}) = \frac{\min_{x' \in \mathbb{A}} \mu^{(n)}(x', 1_m) - \mathbb{E}_n \left[ \min_{x' \in \mathbb{A}} \mu^{(n+q)}(x', 1_m) \mid z^{(n+1:n+q)} = z_{1:q} \right]}{\max_{1 \leq i \leq q} \text{cost}^{(n)}(z^{(n+i)})}, \quad (3.4)$$

We then modify (3.3) by sampling at the batch of points and fidelities that maximize

$$\max_{z_{1:q} \subset \mathbb{A} \times [0,1]^m} \text{q-cfKG}(z_{1:q}) \quad (3.5)$$

Although we have defined the cfKG algorithm's sampling decision theoretically, (3.3) or (3.5) are challenging optimization problems and naive brute-force approaches are unlikely to produce high-quality results with a reasonable amount of computation. Thus, in the next section, we discuss efficient computational methods for solving (3.3) and (3.5).

## 3.3 ENVELOPE-THEOREM-BASED COMPUTATIONAL METHOD

In this section, we describe computational methods for solving (3.3) and (3.5). We describe our method in the context of (3.5), and observe that (3.3) is a special case.

We generalize a recently proposed envelope-theorem based computational method developed for single-fidelity optimization in Wu et al. (2017), which is used to provide unbiased estimators of both q-cfKG and its gradient. We then use stochastic gradient ascent to optimize the q-cfKG acquistion function, optionally with multiple starts.

### 3.3.1 ESTIMATING Q-CFKG

To support computation, we express $\mu^{(n+q)}(x, 1_m)$ that results from a chosen batch of points and fidelities $z^{(n+1:n+q)} = z_{1:q}$ as

$$\mu^{(n+q)}(x, 1_m) = \mu^{(n)}(x, 1_m) + $$
$$K^{(n)}((x, 1_m), z_{1:q}) \left( K^{(n)}(z_{1:q}, z_{1:q}) + \sigma^2 I \right)^{-1} \left( y^{(n+1:n+q)} - \mu^{(n)}(z_{1:q}) \right).$$

Because $y^{(n+1:n+q)} - \mu^{(n)}(z_{1:q})$ is normally distributed with zero mean and covariance matrix $\left(K^{(n)}(z_{1:q}, z_{1:q}) + \sigma^2 I\right)$ with respect to the posterior after $n$ observations, we can rewrite $\mu^{(n+q)}(x, 1_m)$ as

$$\mu^{(n+q)}(x, 1_m) = \mu^{(n)}(x, 1_m) + \tilde{\sigma}_n(x, z_{1:q}) W_q, \tag{3.6}$$

where $W_q$ is a standard $q$-dimensional normal random vector, and

$$\tilde{\sigma}_n(x, z_{(1:q)}) = K^{(n)}((x, 1_m), z_{1:q}) \left(D^{(n)}(z_{1:q})^T\right)^{-1},$$

where $D^{(n)}(z_{1:q})$ is the Cholesky factor of the covariance matrix $K^{(n)}(z_{1:q}, z_{1:q}) + \sigma^2 I$.

Thus, to provide an unbiased Monte Carlo estimator of the expectation within (3.4), we may sample $W_q$, and then calculate $\min_{x' \in \mathbb{A}} \mu^{(n+q)}(x', 1_m) = \min_{x' \in \mathbb{A}} \mu^{(n)}(x, 1_m) + \tilde{\sigma}_n(x, z_{1:q}) W_q$. To do this optimization, we use a second-order continuous optimization method, where the gradient and Hessian of $\mu^{(n+q)}(x, 1_m)$ with respect to $x$ in (3.6) can be computed by calculating the gradient and Hessian of $\mu^{(n)}(x, 1_m)$ and $\tilde{\sigma}_n(x, z_{1:q})$. We can then compute the q-cfKG acquistion function to arbitrary accuracy by averaging many such independent Monte Carlo estimates.

### 3.3.2 Estimating the gradient of q-cfKG

To solve (3.5), we generalize a recently proposed computational method based on the envelope theorem from Wu et al. (2017) to provide an unbiased estimator of the gradient of the q-cfKG acquisition function, and then use stochastic gradient ascent.

Exploiting (3.6), the q-cfKG acquisition function can be expressed as

$$\text{q-cfKG}(z_{1:q}) = \frac{\min_{x \in \mathbb{A}} \mu^{(n)}(x, 1_m) - \mathbb{E}_n\left[\min_{x \in \mathbb{A}}\left(\mu^{(n)}(x, 1_m) + \tilde{\sigma}_n(x, z_{1:q}) W_q\right)\right]}{\max_{1 \leq i \leq q} \text{cost}^{(n)}(x_i, s_i)},$$

where $W_q$ is a standard $q$-dimensional normal random vector, $\tilde{\sigma}_n(x, z_{(1:q)}) = K^{(n)}((x, 1_m), z_{1:q})(D^{(n)}(z_{1:q})^T)^{-1}$, and $D^{(n)}(z_{1:q})$ is the Cholesky factor of the covariance matrix $K^{(n)}(z_{1:q}, z_{1:q}) + \sigma^2 I$. $\nabla\text{q-cfKG}(z_{1:q})$ can be computed from $-\nabla\mathbb{E}_n\left[\min_{x \in \mathbb{A}}\left(\mu^{(n)}(x, 1_m) + \tilde{\sigma}_n(x, z_{1:q}) W_q\right)\right]$ and $\nabla \max_{1 \leq i \leq q} \text{cost}^{(n)}(z^{(n+i)})$, where differentiability of $\text{cost}^{(n)}(\cdot)$ implies $\max_{1 \leq i \leq q} \text{cost}^{(n)}(x_i, s_i)$ is differentiable almost everywhere. To compute the first term, under sufficient regularity conditions (L'Ecuyer, 1990) that we conjecture hold in most applications to hyperparameter tuning, one can interchange the gradient and expectation operators,

$$\nabla\mathbb{E}_n\left[\min_{x \in \mathbb{A}}\left(\mu^{(n)}(x, 1_m) + \tilde{\sigma}_n(x, z_{1:q}) W_q\right)\right] = \mathbb{E}_n\left[\nabla\min_{x \in \mathbb{A}}\left(\mu^{(n)}(x, 1_m) + \tilde{\sigma}_n(x, z_{1:q}) W_q\right)\right]. \tag{3.7}$$

This technique is called infinitesimal perturbation analysis (IPA) (L'Ecuyer, 1990).

Since multiplication, matrix inversion (when the inverse exists), and Cholesky factorization (Smith, 1995) preserve continuous differentiability, $(x, z_{1:q}) \mapsto \left(\mu^{(n)}(x, 1_m) + \tilde{\sigma}_n(x, z_{1:q}) W_q\right)$ is continuously differentiable under mild regularity conditions. When this function is continuously differentiable and $\mathbb{A}$ is compact, the envelope theorem (Milgrom & Segal, 2002, Corollary 4) implies

$$\mathbb{E}_n\left[\nabla\min_{x \in \mathbb{A}}\left(\mu^{(n)}(x, 1_m) + \tilde{\sigma}_n(x, z_{1:q}) W_q\right)\right]$$
$$= \mathbb{E}_n\left[\nabla\left(\mu^{(n)}(x^*(W_q), 1_m) + \tilde{\sigma}_n(x^*(W_q), z_{1:q}) \cdot W_q\right)\right],$$
$$= \mathbb{E}_n\left[\nabla\tilde{\sigma}_n(x^*(W_q), z_{1:q}) \cdot W_q\right],$$

where $x^*(W_q) \in \arg\min_{x \in \mathbb{A}}\left(\mu^{(n)}(x, 1_m) + \tilde{\sigma}_n(x, z_{1:q}) W_q\right)$. We can use this unbiased gradient estimator within stochastic gradient ascent (Harold et al., 2003) to solve the optimization problem (3.5).

### 3.4 EXTENSIONS TO DERIVATIVE-ENABLED SETTINGS

Following Wu et al. (2017), which developed single-fidelity BO methods for use when gradient information is available, we develop a version of the cfKG algorithm, called derivative-enabled cfKG, that can be used when gradients are available in the continuous-fidelity setting. This can be used to reduce the number of function evaluations required to find a high-quality solution to (1.1).

First, observe that a $GP(\mu^n, K^n)$ prior on $g(x, s)$ implies a multi-output $GP(\tilde{\mu}^n, \tilde{K}^n)$ prior on $(g, \nabla_x g)$. Then, observe that the same reasoning we used to develop the cfKG acquistion function in (3.2) can be used when when we observe gradients to motivate the acquisition function,

$$\text{cf-d-KG}(x, s) = \frac{\min_{x \in \mathbb{A}} \tilde{\mu}_1^{(n)}(x, 1_m) - \mathbb{E}_n \left[ \min_{x \in \mathbb{A}} \tilde{\mu}_1^{(n+1)}(x, 1_m) \mid x^{(n+1)} = x, s^{(n+1)} = s \right]}{\text{cost}^{(n)}(x, s)},$$

where $\text{cost}^{(n)}(x, s)$ is now our estimate after $n$ samples of the cost of evaluating both $g$ and its gradient with respect to $x$, and $\tilde{\mu}_1^{(n)}(x, s)$ is the posterior mean on $g(x, s)$ in the multi-output GP. The conditional expectation is taken with respect to both the observed function value and gradient. A batch version of the derivative-enabled cfKG acquistion function can be defined analogously. We use techniques similar to those described in Sect. 3.3 to optimize these acquisition functions.

## 4 NUMERICAL EXPERIMENTS

In this section, we compare sequential, batch, and derivative-enabled cfKG with several benchmark algorithms on four synthetic functions, tuning convolutional neural networks on CIFAR-10 and SVHN, and on large-scale kernel learning. Our benchmarks include the traditional Bayesian optimization algorithms KG (Wu & Frazier, 2016) and EI (Jones et al., 1998), and the multi-fidelity Bayesian optimization with continuous approximation algorithm (BOCA) (Kandasamy et al., 2017). We also compare with Hyperband (Li et al., 2016) in the CIFRA-10 experiment, and with derivative-enabled KG (Wu et al., 2017) in the kernel-learning experiment. We use squared-exponential kernels with constant mean functions and integrate out the GP hyperparameters by sampling $M = 10$ sets of hyperparameters using the `emcee` package (Foreman-Mackey et al., 2013).

### 4.1 OPTIMIZING SYNTHETIC FUNCTIONS

Inspired by numerical experiments in Kandasamy et al. (2017), we modify four standard synthetic test functions, 2-d Branin, 3-d Rosenbrock, 3-d Hartmann, and 6-d Hartmann, by adding 1 fidelity control, as described in detail in Appendix. A. We use cost $\text{cost}(x, s) := \prod_{i=1}^{m}(0.01 + s_i)$ to mimic the linear cost of training a neural network with more data and longer iterations, and $\text{cost}(z_{1:q}) := \max_{1 \leq j \leq q} \text{cost}(z_j)$ to mimic wall-clock time for training $q$ neural networks in parallel. Fig. 1 summarizes the results. cfKG achieves lower simple regret than competitors at the same cost.

### 4.2 TUNING CONVOLUTIONAL NEURAL NETWORKS ON CIFAR-10 AND SVHN

We use sequential and batch cfKG to tune convolution neural networks (CNNs) on CIFAR-10 and SVHN. Our CNN consists of 3 convolutional blocks and a softmax classification layer. Each convolutional block consists of two convolutional layers with the same number of filters followed by a max-pooling layer. There is no dropout or batch-normalization layer. We split the CIFAR-10 dataset into 40000 training samples, 10000 validation samples and 10000 test samples. We split the SVHN training dataset into 67235 training samples and 6000 validation samples, and use the standard 26032 test samples. We apply standard data augmentation: horizontal and vertical shifts, and horizontal flips. We optimize 5 hyperparameters to minimize the classification error on the validation set: the learning rate, batch size, and number of filters in each convolutional block. cfKG and q-cfKG use two fidelity controls: the size of the training set and the number of training iterations. Hyperband uses the size of the training set as its resource (it can use only one resource or fidelity), using a bracket size of $s_{\max} = 4$ as in Li et al. (2016) and the maximum resource allowed by a single configuration set to 40000. We set the maximum number of training epochs for all algorithms to 50 for CIFAR-10 and 40 for SVHN. Fig. 2 shows the performance of cfKG relative to several

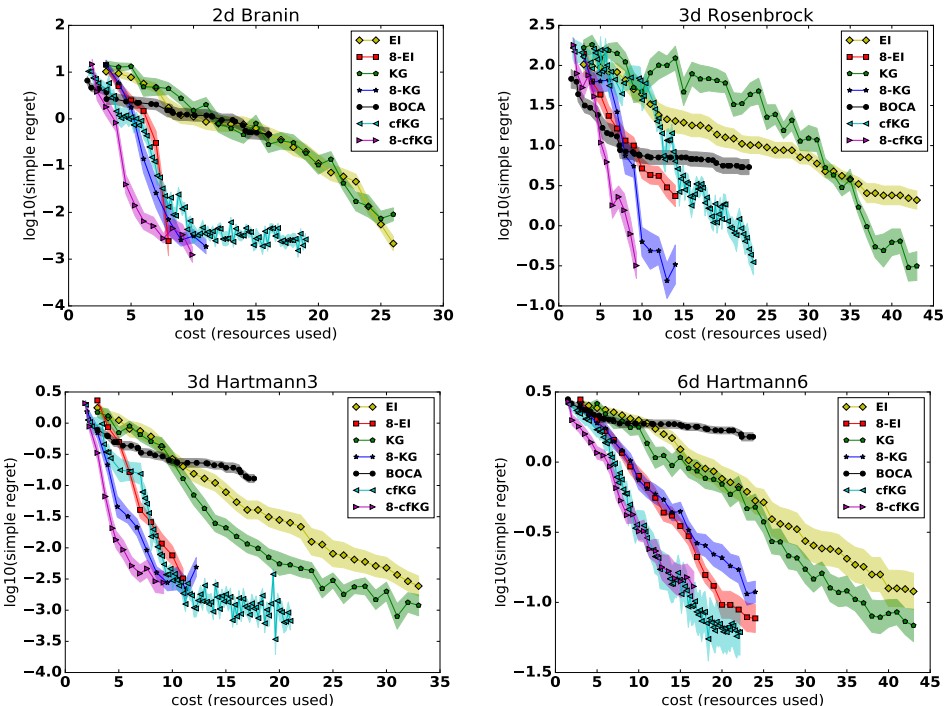

Figure 1: Optimizing 4 synthetic functions: 2-d Branin, 3-d Rosenbrock, 3-d Hartmann, and 6-d Hartmann with 20 independent runs. Fidelity is 1-d.

benchmarks. cfKG successfully exploits the cheap approximations and find a good solution much faster than KG and Hyperband. When we train using optimized hyperparameters on the full training dataset for 200 epochs, test data classification error is $\sim 12\%$ for CIFAR-10 and $\sim 5\%$ for SVHN.

## 4.3 LARGE-SCALE KERNEL LEARNING

We use derivative-enabled cfKG (cf-d-KG) in a large-scale kernel learning example obtained by modifying the 1-d demo example for KISS-GP (Wilson & Nickisch, 2015) on the GPML website (Rasmussen & Nickisch, 2016). In this example, we optimize 3 hyperparameters (the alpha, the length scale and the variance of the noise) of a GP with an RBF kernel on 1 million training points to maximize the log marginal likelihood. We evaluate both the log marginal likelihood and its gradient using the KISS-GP framework. We use two fidelity controls: the number of training points and the number of inducing points in the KISS-GP framework. We set the maximum number of inducing points to $m = 1000$. We compare cf-d-KG to the derivative-enabled knowledge gradient (d-KG) (Wu et al., 2017) both with batch size $q = 8$. Fig. 2 shows that cf-d-KG successfully utilizes the inexpensive function and gradient evaluations to find a good solution more quickly.

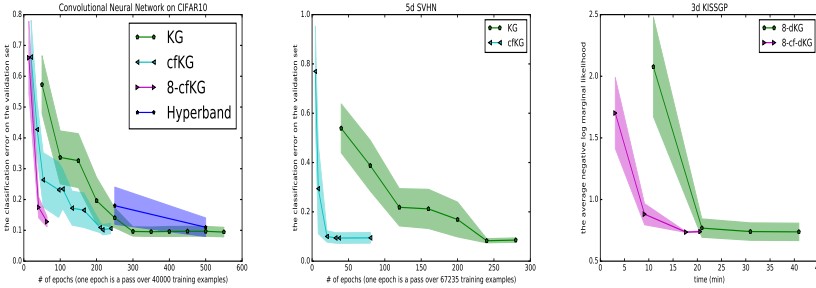

Figure 2: From left to right: Tuning convolutional neural networks on CIFAR-10 and SVHN with 6 independent runs, and kernel learning under the KISS-GP framework with 10 independent runs.

## 5    CONCLUSION

We propose a novel continuous-fidelity BO algorithm, cfKG, which generalizes naturally to batch and derivative settings. This algorithm can find good solutions to global optimization problems with less cost than state-of-art algorithms in applications including deep learning and kernel learning.

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

## A   ADDITIONAL EXPERIMENTAL DETAILS

Here we define in detail the synthetic test functions on which we perform numerical experiments described in Sect. 4.1. The test functions are:

$$
\text{augmented-Branin}(x, s) = \left( x_2 - \left( \frac{5.1}{4\pi^2} - 0.001 * (1 - s_1) \right) x_1^2 + \frac{5}{\pi} x_1 - 6 \right)^2
$$
$$
+ 10 * \left( 1 - \frac{1}{8\pi} \right) \cos(x_1) + 10
$$

$$
\text{augmented-Hartmann}(x, s) = (\alpha_1 - 0.01 * (1 - s_1)) \exp \left( - \sum_{j=1}^{d} A_{ij} (x_j - P_{1j})^2 \right)
$$
$$
+ \sum_{i=2}^{4} \alpha_i \exp \left( - \sum_{j=1}^{d} A_{ij} (x_j - P_{ij})^2 \right)
$$

$$
\text{augmented-Rosenbrock}(x, s) = \sum_{i=1}^{2} \left( 100 * (x_{i+1} - x_i^2 + 0.001 * (1 - s_1))^2 + (x_i - 1)^2 \right).
$$

