# OpenReview forum: "Continuous-fidelity Bayesian Optimization with Knowledge Gradient"
_ICLR.cc/2018/Conference — Reject_

### Official Review · AnonReviewer2 · 2017-11-15
**Neat work of low novelty.**

**Rating:** 5
**Confidence:** 4

**Review:**

This paper studies hyperparameter-optimization by Bayesian optimization, using the Knowledge Gradient framework and allowing the Bayesian optimizer to tune fideltiy against cost.

There’s nothing majorly wrong with this paper, but there’s also not much that is exciting about it. As the authors point out very clearly in Table 1, this setting has been addressed by several previous groups of authors. This paper does tick a previously unoccupied box in the problem-type-vs-algorithm matrix, but all the necessary steps are relatively straightforward.

The empirical results look good in comparison to the competing methods, but I suspsect an author of those competitors could find a way to make their own method look better in those plots, too.

In short: This is a neat paper, but it’s novelty is low. I don't think it would be a problem if this paper were accepted, but there are probably other, more groundbreaking papers in the batch.

Minor question: Why are there no results for 8-cfKG and Hyperband in Figure 2 for SVHN?

---

> ### Author Response · Authors · 2018-01-05
> **Reply to AnonReviewer2**
>
> As in our reply to AnonReviewer3, we would like to emphasize the practical value of a method that effectively leverages continuous fidelities of multiple dimensions (training data size and training iterations) in a batch setting, especially in light of the difficulty of parallelizing other competitive methods in the sequential setting.
>
> In reference to your minor question, we did not add 8-cfKG because (sequential) cfKG already finds an extremely good solution within a single complete training run, and adding parallelism could not improve this.  One can view the performance of 8-cfKG in this example as the same as sequential cfKG.

---

### Official Review · AnonReviewer3 · 2017-11-26
**Incremental Technical Contribution, Weak Empirical Comparisons**

**Rating:** 4
**Confidence:** 5

**Review:**


Many black-box optimization problems are "multi-fidelity", in which it
is possible to acquire data with different levels of cost and
associated uncertainty.  The training of machine learning models is a
common example, in which more data and/or more training may lead to
more precise measurements of the quality of a hyperparameter
configuration.  This has previously been referred to as a special case
of "multi-task" Bayesian optimization, in which the tasks can be
constructed to reflect different fidelities.  The present paper
examines this construction with three twists: using the knowledge
gradient acquisition function, using batched function evaluations, and
incorporating derivative observations.  Broadly speaking, the idea is
to allow fidelity to be represented as a point in a hypercube and then
include this hypercube as a covariate in the Gaussian process.  The
knowledge gradient acquisition function then becomes "knowledge
gradient per unit cost" the KG equivalent to the "expected improvement
per unit cost" discussed in Snoek et al (2012), although that paper
did not consider treating fidelity separately.

I don't understand the claim that this is "the first multi-fidelity
algorithm that can leverage gradients".  Can't any Gaussian process
model use gradient observations trivially, as discussed in the
Rasmussen and Williams book?  Why can't any EI or entropy search
method also use gradient observations?  This doesn't usually come up
in hyperparameter optimization, but it seems like a grandiose claim.
Similarly, although I don't know of a paper that explicitly does "A +
B" for multi-fidelity BO and parallel BO, it is an incremental
contribution to combine them, not least because no other parallel BO
methods get evaluated as baselines.

Figure 1 does not make sense to me.  How can the batched algorithm
outperform the sequential algorithm on total cost?  The sequential
cfKG algorithm should always be able to make better decisions with its
remaining budget than 8-cfKG.  Is the answer that "cost" here means
"wall-clock time when parallelism is available"?  If that's the case,
then it is necessary to include plots of parallelized EI, entropy
search, and KG.  The same is true for Figure 2; other parallel BO
algorithms need to appear.

---

> ### Author Response · Authors · 2018-01-05
> **Reply to AnonReviewer3**
>
> Regarding our claim that this is the first multi-fidelity algorithm to leverage gradients, we searched the literature, and believe that this is indeed true: we were unable to find a paper that uses gradients in a multi-fidelity setting.  At the same time, we do agree that it should be possible to add gradient observations into the inference used by an existing multi-fidelity method, although this is not discussed elsewhere.  For this reason, we have removed this sentence.
>
> We also note that Wu et al. 2017 cited in our paper shows that simply adding gradient observation into GP inference with a standard acquisition function such as EI in the single-fidelity setting is not sufficient to provide a substantial performance improvement over the setting without gradient observations.  It is important to additionally modify the acquisition function to sample at points where gradient observations are particularly helpful.  For this reason we suspect that our method would outperform an existing multi-fidelity method whose inference but not acquisition function was modified to use gradients.  Since we do not do numerical experiments to confirm this fact, we do not discuss it in the paper.
>
> Regarding our claim that this is the first parallel multi-fidelity method, our response is similar: we believe this is true, as we searched the literature and did not find an existing paper that does this, but we do agree that this point does not need to be emphasized, and so we removed the sentence that claimed it.  Regarding the absence of parallel BO baselines, we have corrected this and now have two parallel BO baselines in our synthetic experiments as discussed below.
>
> At the same time, when thinking about parallelizing an existing multi-fidelity method based on expected improvement or entropy search, ES tends to outperform EI in multi-fidelity settings, and ES is challenging to parallelize.  As far as we know, [1] is the only paper to do so, and its method incurs significant computational cost with no code publicly available.  It is perhaps for this reason that we were unable to find any previous papers or any publicly available software that supported multi-fidelity BO with batch evaluations, despite the apparent practical importance of this problem class.  Moreover, Hyperband is also difficult to parallelize due to its communication overhead, as discussed by the authors when leaving parallelization to future work.  Thus, we view developing an effective parallel multi-fidelity method, as we have done in this paper, as an important contribution.
>
> Regarding Figure 1:
> Yes, cost here is wall-clock time.  Thus, batched algorithms tend to have smaller wall-clock times than sequential algorithms.  As per your suggestion we have added two additional parallel BO benchmarks: parallel EI and parallel KG.  We include them in Figure 1 in the revised version of the paper.  Batch cfKG outperforms both of these batch benchmarks.
>
> [1] Shah, Amar, and Zoubin Ghahramani. "Parallel predictive entropy search for batch global optimization of expensive objective functions." Advances in Neural Information Processing Systems. 2015.

---

> > ### Author Response · Authors · 2018-01-05
> > **Continued**
> >
> > In addition, in response to these questions about EI and ES, we have added a brief discussion of EI and ES in comparison with KG, and its level of appropriateness for multi-fidelity optimization, in the introduction.

---

### Official Review · AnonReviewer1 · 2017-11-27
**a paper with some new results**

**Rating:** 6
**Confidence:** 5

**Review:**

Minor comments:
- page 7. “Then, observe that the same reasoning we used to develop the cfKG acquistion function
in (3.2) can be used when when we observe gradients to motivate the acquisition function…” - some misprints, e.g. double “when”
- The paper lacks theoretical analysis of convergence of the proposed modification of the knowledge gradient criterion.

Major comments:

Current approaches to optimisation of expensive functions are mainly based on Gaussian process model. Such approaches are important for Auto ML algorithms.

There are a lot of cases, when for an expensive function we can obtain measurements of its values with continuous fidelity by leveraging costs for evaluation vs. fidelity of the obtained values. E.g. as fidelity we can consider a size of the training set used to train a deep neural network.

The paper contains a some new algorithm to perform Bayesian optimisation of a function with continuous fidelity. Using modification of the knowledge gradient acquisition function the authors obtained black box optimisation method taking into account continuous fidelity.

Due to some reason the authors forgot to take the cost function into account when formulating the algorithm 1 in 3.3.2 and corresponding formula (3.7).

So, the logic of the definition of q-cfKG is understandable, but the issue with the missing denominator, containing cost function, remains.

The approach, proposed in section 3.3.2, looks as follows:
- the authors used formulas from [Wu t al (2017) - https://arxiv.org/abs/1703.04389]
- and include additional argument in the mean function of the Gaussian process.
However, in Wu t al (2017) they consider usual knowledge gradient, but in this paper they divide by the value of max(cost(z)), which is not differentiable.

Other sections of the paper are sufficiently well written, except
- the section 3.3.2,
- section with results of experiments: I was not able to understand how the authors defined cost function in sections 4.2 and 4.3 for their neural network and large scale kernel learning.

In principle, the paper contains some new results, but it should be improved before publishing.

---

> ### Author Response · Authors · 2018-01-05
> **Reply to AnonReviewer1**
>
> Thanks for pointing out the mistake in formula (3.7). We have not made the error in the implementation and the experiments are valid. We have corrected it in the revised version of the paper.  This should make section 3.3.2 more clear.
>
> The cost function max_{1 <= i <= q} (cost(z_i)) is differentiable almost everywhere if cost(z) is differentiable everywhere.  The points of non-differentiability are those points where there are ties in the maximum.  Because they have measure 0, and our stochastic gradient estimator has continuous support (when the predictive distribution at the proposed points to sample is not degenerate), our stochastic gradient ascent algorithm encounters these points with probability 0 as long as it does not start at such a point.  We now discuss this in the revised version of the paper.
>
> We take the cost function for tuning a neural network in section 4.2 and 4.3 to be the number of training examples used during the training process divided by the original number of training examples.  For example, if we subsample 10,000 training points (out of 50,000) per epoch, and train with 20 epochs, then the cost is 10,000*20/50,000 = 4. This definition is analogous to the resource R in the Hyperband paper. We take the cost function for kernel learning in the synchronous setting to be the wall-clock time. In batch settings, we take the cost to be max_{1 <= i <= q} (cost(z_i)), modeling the wall-clock that one would have when running jobs in parallel.

---

### Decision · Program_Chairs · 2018-01-29
**ICLR 2018 Conference Acceptance Decision**

**Decision:**

Reject

**Comment:**

This paper combines multiple existing ideas in Bayesian optimization (continuous-fidelity, use of gradient information and knowledge gradient) to develop their proposed cfKG method.  While the methodology seems neat and effective, the reviewers (and AC) found that the presented approach was not quite novel enough in light of existing work to justify acceptance to ICLR.  Continuous fidelity Bayesian optimization is well studied and knowledge gradient + derivative information was presented at NIPS.  The combination of these things seems quite sensible but not sufficiently novel (unless the empirical results were *really* compelling).

Pros:
- The paper is clear and writing is of high quality
- Bayesian optimization is interesting to the community and compelling methods are potentially practically impactful
- Outperforms existing methods on the chosen benchmarks

Cons:
- Is an incremental combination of existing methods
- The paper claims too much